# LymphoTrack Is Equally Sensitive as PCR GeneScan and Sanger Sequencing for Detection of Clonal Rearrangements in ALL Patients

**DOI:** 10.3390/diagnostics12061389

**Published:** 2022-06-04

**Authors:** Karin Paulsen, Millaray Marincevic, Lucia Cavelier, Peter Hollander, Rose-Marie Amini

**Affiliations:** 1Clinical and Experimental Pathology, Department of Immunology, Genetics and Pathology, Uppsala University, 75185 Uppsala, Sweden; kapaulsen82@gmail.com (K.P.); millaray.marincevic.zuniga@akademiska.se (M.M.); rose-marie.amini@igp.uu.se (R.-M.A.); 2Science for Life Laboratory, Department of Immunology, Genetics and Pathology, Uppsala University, 75236 Uppsala, Sweden; lucia.cavelier.franco@ki.se

**Keywords:** LymphoTrack, NGS, GeneScan, rearrangement, Ig genes, TCR genes, BIOMED2, MRD, minimal measurable disease, acute lymphoblastic leukemia

## Abstract

Monoclonal rearrangements of immunoglobulin (Ig) genes and T-cell receptor (TCR) genes are used for minimal measurable disease in acute lymphoblastic leukemia (ALL). The golden standard for screening of gene rearrangements in ALL has been PCR GeneScan and Sanger sequencing, which are laborsome and time-consuming methods. More rapid next-generation sequencing methods, such as LymphoTrack could possibly replace PCR GeneScan and Sanger sequencing for clonality assessment. Our aim was to evaluate to what extent LymphoTrack can replace PCR GeneScan and Sanger sequencing concerning sensitivity and quantifiability in clonality assessment in 78 ALL samples. With LymphoTrack, clonality assessment was based on the %Total reads, where ≥10% was used as cut off for clonal rearrangements. The patients displayed 0 to 4 clonal rearrangements per assay. The detection rate (rearrangements detected with PCR GeneScan and/or Sanger sequencing, also detected with LymphoTrack) was 85/85 (100%) for IGH, 64/67 (96%) for IGK, 91/93 (98%) for TCRG and 34/35 (97%) for TCRB. Our findings demonstrate that LymphoTrack was equally sensitive in detecting clonal rearrangements as PCR GeneScan and Sanger Sequencing. The LymphoTrack assay is reliable and therefore applicable for clonal assessment in ALL patients in clinical laboratories.

## 1. Introduction

In acute lymphoblastic leukemia (ALL), unique monoclonal rearrangements of immunoglobulin genes (Ig) and/or T-cell receptor genes (TCR) are identified and used in clinical diagnostics as clonotypic markers for minimal residual/measurable disease analyses (MRD) [1,2,3,4,5,6]. Ig and TCR rearrangements are used as templates for design of allele specific oligonucleotides (ASO) and are targets for MRD because of their high diversity [7,8]. Monitoring MRD with quantitative polymerase chain reaction (qPCR) is the current golden standard and requires prior knowledge of patient-specific Ig and TCR rearrangements [9,10]. It is therefore of utmost importance that methods used for clonal assessment are sensitive and generate accurate and reliable sequences.

During the last decade, the golden standard for screening of clonal rearrangements in ALL has been PCR GeneScan and Sanger sequencing, which are labor-intensive methods [11,12]. Hence, methods that require less hands-on time and have higher throughput of high-quality data are desired. Compared to the previous gold standard method of sanger sequencing, which sequences a single DNA fragment at a time, molecular diagnostics has moved towards next-generation sequencing (NGS) technology, the processing of millions of fragments simultaneously by deeper sequencing depth. Recent advances in NGS technology [13], combined with detailed knowledge of current primer systems for Ig and TCR genes have allowed the development of LymphoTrack, a relatively rapid, reliable, and affordable assay that may be applicable in routine practice for clonality assessment of Ig and TCR genes. The high-level multiplexing capability of NGS allows design of families of balanced primers that simultaneously amplify all possible combinations of the rearranged Ig and/or TCR loci [14].

Whether LymphoTrack has the power to replace GeneScan for detection of clonal rearrangements in ALL samples in a clinical setting is still not fully elucidated [15,16]. Thus, the aim of this study was to compare the performance of LymphoTrack Dx Assay Panels protocol (InVivoScribe Technologies, Inc. San Diego, CA, USA) compared to GeneScan and Sanger sequencing, using PCR-based BIOMED2 and IdentiClone Clonality Assays (InVivoScribe) primers [17] to detect clonal Ig and TCR rearrangements in a single center cohort consisting of 78 ALL patients.

## 2. Materials and Methods

A total of 78 bone marrow (BM) samples from patients diagnosed with ALL at the Uppsala University Hospital were evaluated (56 B-ALL and 22 T-ALL). The age distribution among the patients was between 1 and 69 years. Tumor blast count ranged from 30 to 98% after Ficoll separation. Control DNA from tonsils provided from the LymphoTrack kit was used as control material in this study. This study was approved by the Regional Ethical Review Board in Uppsala, Sweden (EPN 2014/233) and the study was performed in accordance with the Declaration of Helsinki.

### 2.1. Sample Preparation

Ficoll-isopaque-gradient separation of mononuclear cells was performed according to manufacturer instructions prior to freezing the cells in DMSO in liquid nitrogen. Isolation of DNA was performed using the Qiamp^®^ DNA Blood Mini Kit (Qiagen, Hilden, Germany) and DNA concentration was measured using the NanoDrop 2000 system (Thermo Fisher Scientific Inc., Waltham, MA, USA). All DNA samples were diluted to 70 ng/µL for optimal downstream applications.

### 2.2. PCR Amplification and GeneScan Analysis

PCR reactions, for complete and/or incomplete Ig and TCR rearrangements, were performed using BIOMED2 primers and IdentiClone^®^ Clonality Assays for ABI Fluorescence detection (InVivoScribe Technologies, San Diego, CA, USA) (Table 1). AmpliTaq Gold™ DNA Polymerase (Thermo Fisher Scientific, Vilnius, Lithuania) and HotstarTaq^®^ Plus Master Mix (Qiagen, Hilden, Germany) were used as enzymes for the IdentiClone^®^ Clonality Assays and BIOMED2 protocol respectively.

Fragment length of the fluorescently labeled PCR products was assessed using automated capillary gel electrophoresis on the 3500 Genetic Analyzer (Applied Biosystems™, Foster City, CA, USA). Product sizes were calibrated with Gene Scan™ 500 ROX™ Size Standard (Life Technologies, Warrington, UK). Furthermore, fragment analysis was performed using GeneMapper Software v5 (Applied Biosystems™, Foster City, CA, USA) [18]. Interpretation of clonal rearrangements based on intensity of generated peaks for IGH, IGK, TCRG and TCRB was performed according to current guidelines [19]. Furthermore, PCR-products including amplicons fulfilling the criteria for clonality were subjects for Sanger sequencing.

Prior to Sanger sequencing, PCR products were purified with Exonuclease I (Thermo Fisher Scientific, Carlsbad, CA, USA) and Shrimp Alkaline Phosphatase (GE Healthcare UK Ltd., Amersham, UK). BIOMED2 primers and BigDye Terminator v1.1 Cycle sequencing Kit (Thermo Fisher Scientific, Vilnius, Lithuania) were used for the sequencing reaction setup. The DNA fragments from the sequencing reaction were purified using BigDye^®^Xterminator™ Purification kit (Life Technologies™, Framingham, MA, USA) and furthermore, separated using capillary electrophoresis on a 3500 Genetic Analyzer (Applied Biosystems™, Foster City, CA, USA). Detected sequences were analyzed using Sequencher 5.4 Software (Gene Codes, Ann Arbor, MI, USA). Obtained sequences were submitted to IMGT/V-QUEST Junction analysis tool in order to determine the specific gene families involved in the rearrangement and for analysis of the junctional region [20]. Annotations of the respective genes are presented in Appendix A.

### 2.3. Next Generation Sequencing

LymphoTrack^®^ Dx Assay Panels used in this study were LymphoTrack^®^ Dx IGH FR1 Assay Panel-MiSeq^®^ (B-ALL), LymphoTrack^®^ Dx IGK Assay Panel-MiSeq^®^ (B-ALL), LymphoTrack^®^ Dx TRG Assay Panel-MiSeq^®^ (B- and T-ALL) and LymphoTrack^®^ Dx TRB Assay Panel-MiSeq^®^ (T-ALL) (Table 1). Each assay panel consists of 24 Illumina indexed master mixes for multiplex PCR. Prior to NGS library preparation, DNA concentration was measured using Qubit™ dsDNA BR Assay Kit (Invitrogen by Thermo Fisher Scientific, Waltham, MA, USA, Life Technologies™, Eugene, OR, USA) on Qubit^®^ 2.0 fluorometer (Invitrogen by Thermo Fisher Scientific, Waltham, MA, USA). LymphoTrack^®^ Assay Panels require a minimum of 50 ng input DNA to each amplicon reaction. However, we aimed for 150 ng to facilitate downstream quantification steps, hence the concentration of each DNA sample was adjusted to 30 ng/µL prior to PCR setup. Furthermore, 5 µL of the diluted DNA was added to 45 µL LymphoTrack index master mix and 0.2 µL Eagle Taq DNA Polymerase (Roche). Downstream of the PCR, amplicons were purified using AMPureXP purification beads (Beckman coulter, Brea, CA, USA). Subsequently, each amplicon was quantified on the 7500 Fast Real Time PCR System (Applied Biosystems by Life Technologies™, Eugene, OR, USA, Thermo Scientific, Waltham, MA, USA) using KAPA Library Quantification Kit for Illumina (Roche). All amplicons from each assay (IGH FR1, IGK, TRG and TRB respectively) were pooled in equimolar amounts into one library per assay. Each assay library pool was in turn quantified with qPCR and multiplexed into one final library of 4 nM. The final library pool was loaded onto the MiSeq Sequencing system (Illumina). Different MiSeq sequencing kits were used depending on the library size and number of samples: MiSeq Reagent Nano Kit v2, 500 cycles (Illumina) for ≤4 samples; MiSeq Reagent Kit v2, 500 cycles (Illumina) for 5–8 samples.

The LymphoTrack Dx software–MiSeq Version 2.4.3 (InVivoScribe) was used to analyze raw sequence data in FASTQ format. For each assay, the Merged Read Summary report was applied for identification of the top merged sequences and the frequency of these, in order to assess clonality. The frequency of identical sequences for each sample and assay was calculated by dividing the number of identical sequence reads by the total number of sequence reads in the sample (Figure 1).

To streamline the interpretation of the LymphoTrack results, we established interpretation criteria applicable for all LymphoTrack assays, based on the IVS’ interpretation criteria in each of the assay IFUs combined with the patterns demonstrated among the LymphoTrack results in this study (Figure 2). For clonality assessment, a 10% frequency was determined as the cutoff for defining a leukemic clone, provided that the number of merged reads was ≥20,000. For rearrangements resulting in 10,000–20,000 merged reads and between 5 and 10% of the total reads, each case was reviewed individually. All of the cutoffs were based on the LymphoTrack kit guidelines. The polyclonal pattern was assigned when the interpretation criteria %Total reads and/or merge count were not fulfilled.

### 2.4. Comparison between GeneScan and Sanger Sequencing with LymphoTrack

Clonal rearrangements detected with PCR GeneScan and Sanger sequencing or PCR GeneScan only were compared to the rearrangements detected with LymphoTrack. Comparison of sequences generated by Sanger sequencing and LymphoTrack was performed on at least 10 samples for each target to confirm the sensitivity of LymphoTrack. When 100% sequence identity was achieved between the methods, the results from PCR GeneScan were used to confirm the LymphoTrack results.

### 2.5. Comparison between RFU and %Total Reads

To compare the degree of correlation between the methods, the rearrangement that generated the highest relative fluorescent unit (RFU) value in PCR GeneScan was compared to the most frequent amplified sequence in LymphoTrack, which consequently generated the highest %Total reads. To obtain an accurate comparison, samples with only one rearrangement were excluded, hence only samples with two or more clonal rearrangements were compared. In samples where PCR GeneScan demonstrated two rearrangements of the same gene family, the highest peak in PCR GeneScan was expected to correspond to the rearrangement with the highest %Total reads in LymphoTrack.

## 3. Results

A comparison between detected clonal rearrangements was performed where the number of clonal rearrangements ranged from 0 to 4 among the assays. An expected detection rate of at least 95% was established with the LymphoTrack assay. However, analysis of data demonstrating as high as 100% correlation for certain rearrangements was also achieved.

### 3.1. IGH

Of 85 rearrangements detected with PCR GeneScan, LymphoTrack detected 85 (100%). However, nine (11%) of the 85 rearrangements did not pass the interpretation criteria for LymphoTrack (Figure 3). Additional investigation of the nine peaks detected using PCR GeneScan demonstrated low peaks or peaks in a polyclonal background. Generally, such rearrangements are not used for downstream applications due to challenges in Sanger sequencing [17]. Polyclonal patterns were observed with PCR GeneScan in six samples and furthermore confirmed with LymphoTrack (Appendix A). One discrepancy was detected in sample 10 where the LymphoTrack Software assessed the most frequent rearrangement to VH3-30–JH5 whilst IMGT/V-QUEST assessed the same sequence to VH3-30–JH4. A confirming PCR with the BIOMED2 IGHB tube and Sanger sequencing was performed, followed by analysis with IMGT/V-QUEST and manual interpretation using hard copies of the JH-genes, which resolved the involvement of a JH4 gene in the rearrangement. In this study, investigation of clonal compartments of IGH gene rearrangements in immature lymphoblasts was of interest and therefore only the FR1 region was investigated. Moreover, looking at the concordance ratio of detected IGH genes between NGS and GeneScan, a satisfying level was reached, emphasizing the use of only the FR1 tube.

### 3.2. IGK

Of 67 rearrangements identified with PCR GeneScan, the LymphoTrack assay could detect 64 of them (96%). However, five (7%) of the 67 rearrangements did not pass the interpretation criteria for LymphoTrack (Figure 3). Additional investigation of the five peaks in PCR GeneScan demonstrated low peaks or peaks in polyclonal background with one exception, in sample 55; PCR GeneScan demonstrated a significantly high peak for an Intron-Kde rearrangement, which accounted for only 6.2% of the total reads in LymphoTrack. The rearrangement was later confirmed with Sanger sequencing. Polyclonal patterns were observed with PCR GeneScan in 21 samples and were confirmed with LymphoTrack with one exception. In sample 54, PCR GeneScan demonstrated a polyclonal pattern with the KA tube, whilst LymphoTrack detected a Vκ-Jκ rearrangement, representing 24.5% of the total reads (Appendix A).

### 3.3. TRG

Of 93 rearrangements detected with PCR GeneScan, LymphoTrack detected 91 (98%). However, four (4%) of the 93 rearrangements detected with LymphoTrack did not pass the interpretation criteria (Figure 3). For example, revision of sample 12 showed a small peak with a polyclonal background in PCR GeneScan. Sample 25 showed a significantly high peak for a Vγ9 rearrangement, which accounted for only 2.4% of the total reads in LymphoTrack. Subsequently, this Vγ9 rearrangement was confirmed with Sanger sequencing. In sample 49, PCR GeneScan demonstrated a low peak in a polyclonal background, which could not be confirmed with Sanger sequencing. In sample 56, the peak corresponding to the Vγ3 rearrangement is small, which correlates with the low number of detected % total reads.

In two samples (43 and 67), LymphoTrack displayed polyclonal patterns, while PCR GeneScan detected a minor Vγ9 peak in a polyclonal background, which was also confirmed with Sanger sequencing. 

LymphoTrack detected seven Vγ11 rearrangements (ORF) in five samples (3, 4, 20, 45 and 51) where PCR GeneScan demonstrated a polyclonal pattern with the BIOMED2 primers (GA- and GB tube). To confirm the findings in LymphoTrack, a PCR reaction for each of the five samples was performed using the IdentiClone^®^ TCRG Gene Rearrangement Assay 2.0 for PCR GeneScan (IVS-1T-1F, Released 2019). In four of the five samples, PCR GeneScan demonstrated a significantly high peak for Vγ11, however sample 51 remained polyclonal. Due to these findings, ASOs for qPCR were designed, targeting the junctional region of the detected Vγ11 rearrangements [17]. Amplification curves, passing defined criteria, were detected, hence verifying all seven Vγ11 rearrangements.

With PCR GeneScan, polyclonal patterns were observed in 24 samples and furthermore confirmed with LymphoTrack with four exceptions. In samples 4, 45 and 51, LymphoTrack detected 1 or more Vγ11 rearrangements, as mentioned above. In the fourth sample (sample 30), LymphoTrack detected a Vγ2 rearrangement with 89.4% total reads. A more thorough analysis of the Vγ2 sequence retrieved from LymphoTrack, using IMGT/V-QUEST, revealed a significant number of nucleotides that had been trimmed off the Jγ-gene. Heavy trimming of the Jγ-gene may impede successful binding of the BIOMED2 Jγ-primers to its designated primer site, hence the absence of amplification of the rearrangement in PCR GeneScan. Furthermore, PCR was performed using the IdentiClone^®^ TCRG Gene Rearrangement Assay 2.0 for PCR GeneScan (IVS-1T-1F) confirming the Vγ2 rearrangement. (Appendix A).

### 3.4. TRB

Of 35 rearrangements detected with PCR GeneScan, LymphoTrack detected 34 (97%). However, one (3%) of the 34 rearrangements detected with LymphoTrack did not pass the interpretation criteria (Figure 3). Additional investigation of this peak found in sample 73 revealed a peak with a relatively low RFU as detected by PCR GeneScan, which may explain the low amplification in LymphoTrack.

Furthermore, in sample 61, PCR GeneScan demonstrated a minor peak representing a Vβ-Jβ1 rearrangement, which was not amplified with LymphoTrack. Hence, the Sanger sequencing generated a sequence of poor quality, which could not be recognized by IMGT/V-QUEST. In sample 63, a Vβ14-Jβ1.1 rearrangement was detected with LymphoTrack, which corresponded to a Vβ-Jβ1 rearrangement displayed in PCR GeneScan. However, the Vβ14 gene could not be confirmed using Sanger sequencing, nevertheless when using ASO for qPCR, the clone could be confirmed. Polyclonal patterns were observed with PCR GeneScan in three samples and were confirmed with LymphoTrack. In sample 58, LymphoTrack and PCR GeneScan detected three Vβ-Jβ rearrangements where only two could be confirmed with Sanger Sequencing (Appendix A). In sample 58, the LymphoTrack Software assessed the third most frequent rearrangement to a Vβ6-3, whilst the IMGT/V-QUEST database stated “Vβ6-2 or Vβ6-3” for the sequence. Analysis of the Vβ6-3 sequence retrieved from Sanger sequencing in the IMGT/V-QUEST database also resulted in “Vβ6-2 or Vβ6-3”.

### 3.5. Correlation between RFU and %Total Reads in Samples with Two or More Rearrangements

In 26 of 28 samples (93%) for IGH, 15 of 19 samples (79%) for IGK, 26 of 41 (63%) samples for TRG, and 7 of 13 samples (54%) for TRB, the rearrangement that generated the highest RFU value correlated with the rearrangement that generated the highest %Total reads (Figure 4). These findings are further described in the supplementary results.

### 3.6. Ambiguous Rearrangements

A subset of the peaks that were generated with the IVS IGK and TRB tubes for PCR GeneScan appeared to involve more than one clonal rearrangement. Prior to NGS, we would have to clone to resolve these cases; one of the main advantages of NGS is that time consuming cloning is no longer required. While PCR GeneScan detected one single peak in sample 13 and 78, LymphoTrack detected several clonal rearrangements of the same family. In sample 13, PCR GeneScan detected one Vκ–Jκ rearrangement while LymphoTrack revealed two sequences of the same length, Vκ1-33–Jκ4 and Vκ1-43–Jκ2. In sample 78, PCR GeneScan detected one Dβ1–Jβ1 rearrangement while LymphoTrack revealed two different sequences (Dβ1–Jβ1.1 and Dβ1–Jβ1.2) with only a 10 bp difference (Appendix A). There were also cases where PCR GeneScan detected several peaks while LymphoTrack generated one single sequence. In sample 8, 14 and 52, two distinct Vκ7 peaks were detected with PCR GeneScan while only one Vκ7-3–Kde sequence was detected with LymphoTrack (Appendix A).

## 4. Discussion

In this study, we evaluated if the BIOMED2 and InVivoScribe protocols used for PCR GeneScan and Sanger sequencing may be replaced with the streamlined NGS method LymphoTrack for sequence-based screening of clonal rearrangements at initial diagnosis in ALL. We found high detection rates at 96–100% for the LymphoTrack assays IGH FR1, IGK, TRG and TRB compared to the corresponding assays for PCR GeneScan regarding clonal assessment. Thus, using LymphoTrack to detect clonal rearrangements, suitable for MRD follow-up in ALL patients, seems reasonable.

### 4.1. Benefits of LymphoTrack for MRD Screening in ALL

Each LymphoTrack Assay contains one single tube, which makes the NGS workflow much more streamlined and allows higher throughput. LymphoTrack is a robust method that allows a wide concentration range of amplicons from 0.4 to 240 nM, which after pooling and sequencing generates correct sequences. The turnaround time was substantially reduced from approximately 2 weeks using PCR GeneScan and Sanger sequencing down to 3–5 days using LymphoTrack. Additionally, the sequencing time and cost can be heavily reduced by using the Illumina MiSeq v2 500 Nano kit for ≤4 samples, which takes only 24 h on the MiSeq compared to 39 h with the Illumina MiSeq v2 500 kit.

### 4.2. Considerations When Implementing LymphoTrack in MRD Screening in ALL

In this study, a total of seven cases (9%) were found to only have one rearrangement detected with LymphoTrack. According to the ESHLO guidelines, the use of two rearrangements are recommended as MRD targets [21]. Incomplete IGH and/or TCRD rearrangements are common in ALL, however the currently available LymphoTrack assays do not allow for detection of incomplete IGH or TCRD rearrangements [22,23,24,25]. In T-ALL, TCR gene rearrangements are stable, particularly TCRD gene rearrangements show high stability [25]. Thus, using LymphoTrack as the sole method for clonality assessment with available kits might seem inadequate and complementary analyses with GeneScan and Sanger sequencing for incomplete IGH and TCRD rearrangements may be needed in some cases.

### 4.3. The LymphoTrack Software

The LymphoTrack Software is a streamlined and user-friendly bioinformatics tool, using FASTQ files for demultiplexing and sorting of generated sequences by frequency in a comprehensive report in pdf format. For shorter reads generated by LymphoTrack, the LymphoTrack software often failed to identify the J-gene involved in the rearrangement, which was displayed as “none” in the Merged read summary report. Additionally, in a few samples (10 (IGH) and 58 (TRB)) the LymphoTrack software stated a different gene family than the IMGT database. To prevent overlooking any important information about the sequences generated by the LymphoTrack software, analysis using the IMGT/V-QUEST tool, or hard copies of the genes, is strongly recommended since the LymphoTrack Software only states one gene family per column.

### 4.4. Polyclonal Patterns

The methods were consistent regarding polyclonal patterns, with only a few exceptions. In sample 54 (IGK) and sample 30 (TRG) where PCR GeneScan detected polyclonal patterns, LymphoTrack detected clonal rearrangements (Appendix A). The absence of amplification of clonal rearrangement in the KA tube for one sample (54) could not be explained although deviations in the primer systems could be one suggestion. In one sample (30), the absence of amplification in the GA and/or GB tube was due to a seemingly large, trimmed region in the Jγ-gene, which altered the priming binding site for the BIOMED2 primers. Very occasionally, some rearrangements were resolved with Sanger sequencing even though it was displayed as a minor peak in a polyclonal background while LymphoTrack displayed a polyclonal pattern (e.g., in sample 43 and 67 (TRG)). However, since the peaks were of very low significance from a clonality assessment aspect, the results from LymphoTrack were considered more reliable.

### 4.5. Minor Peaks with Background

Minor peaks in a polyclonal background in PCR GeneScan did not seem to affect the amplification efficiency in LymphoTrack since these rearrangement patterns did not pass the interpretation criteria for LymphoTrack. In addition, GeneScan did not seem to affect the amplification efficiency in LymphoTrack. However, to evaluate the absence of amplification in LymphoTrack, such as overseeing clonal rearrangements suitable for MRD, extended Sanger sequencing would be required. Contrariwise, PCR GeneScan generated a major peak while the %Total reads in LymphoTrack did not pass the interpretation criteria in a few samples (36, 55 (IGK) and 25 (TRG)), which is inexplicable.

### 4.6. Correlation between RFU and %Total Reads in Samples with Two or More Rearrangements

In samples with two or more rearrangements detected by PCR GeneScan and NGS LymphoTrack, the correlation between the assays to identify the dominant rearrangement varied from 54% to 93%. The seemingly low correlations for the TRG and TRB assays are nevertheless satisfactory since using the major peak in PCR GeneScan for Sanger sequencing does not always generate a sequence of the junctional region suitable for optimal ASO design. The ASO might not be specific enough, or may not pass defined criteria for MRD markers. Overall, LymphoTrack was a more sensitive method than PCR GeneScan and Sanger sequencing. Samples containing several rearrangements belonging to a single gene family (e.g., sample 58 and 63 (TRB)) usually require additional time-consuming methods such as Homo/Hetero duplex gel and/or cloning to receive the same resolution with PCR GeneScan and Sanger sequencing [18]. With LymphoTrack Software, the sequences generated from MiSeq are visualized in a comprehensible report after demultiplexing and sorting on sequence frequency.

## 5. Conclusions

In conclusion, the NGS-based clonality assay LymphoTrack revealed a high sensitive detection rate of clonal rearrangements in diagnostic samples of ALL as the conventional PCR GeneScan method. There was a satisfactory level of concordance between the methods regarding the detection of the most frequently amplified rearrangement. The LymphoTrack software, where both experimental and bioinformatics analysis processes were streamlined, was advantageous regarding turnaround time. Together, these results highlight the power and benefits of LymphoTrack NGS assay and may serve as a powerful screening method for clonal assessment in ALL patients in clinical laboratories. We suggest that the LymphoTrack NGS assay can be applicable for clonal assessment in ALL patients in clinical laboratories.

## Figures and Tables

**Figure 1 diagnostics-12-01389-f001:**
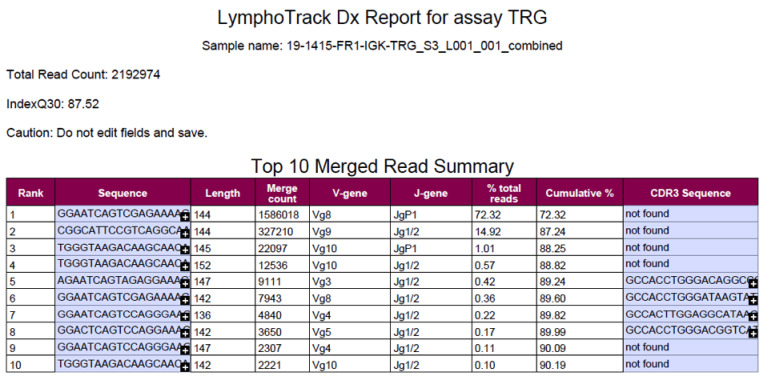
The frequency of identical sequences for each sample and assay is calculated by dividing the number of identical sequence reads by the total number of sequence reads in the sample.

**Figure 2 diagnostics-12-01389-f002:**
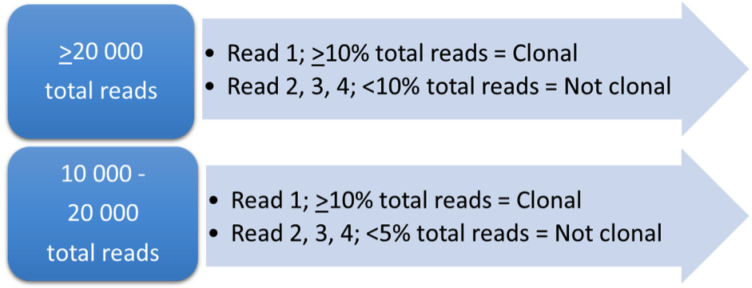
Interpretation criteria based on IVS IFUs for LymphoTrack and the results in this study. For rearrangements resulting in <20,000–10,000 total reads and between 5 and 10% of the total reads, each case was reviewed individually.

**Figure 3 diagnostics-12-01389-f003:**
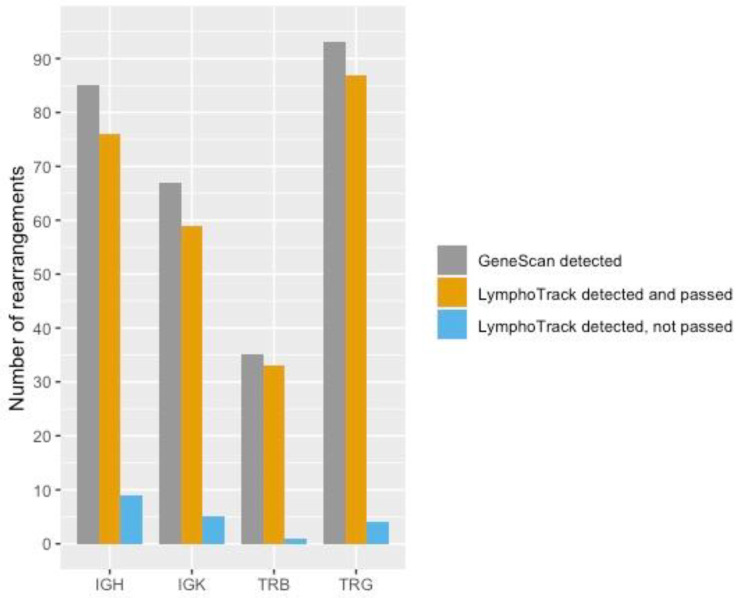
Number of detected rearrangements for each LymphoTrack assay and PCR GeneScan and number of rearrangements detected with LymphoTrack but that did not pass the current interpretation criteria.

**Figure 4 diagnostics-12-01389-f004:**
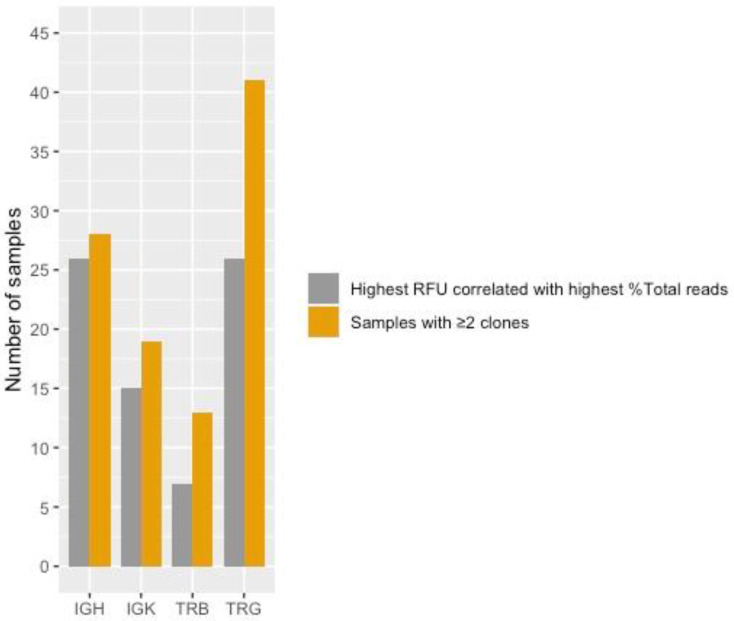
Number of samples with two or more clones for each assay and number of samples where the rearrangement with the highest RFU (detected with PCR GeneScan) correlated with the rearrangement with the highest %Total reads (detected with LymphoTrack).

**Table 1 diagnostics-12-01389-t001:** Primer assays used for the different targets. Assays in parentheses were only used for verification of separate TCRG rearrangements.

B-ALL Primer Assays	IGH (V-D-J)	IGK	TCRG	
BIOMED2 primers	Tube A, B	Tube A	Tube A, B	
IVS IdentiClone	-	Tube B	(G 2.0)	
LymphoTrack	IGH FR1	IGK	TRG	
**T-ALL Primer Assays**			**TCRG**	**TCRB**
BIOMED2 primers			Tube A, B	-
IVS IdentiClone			(G 2.0)	Tube A, B, C
LymphoTrack			TRG	TRB

## Data Availability

The datasets used and/or analyzed during the current study are available on reasonable request.

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
