# Peer review of "LymphoTrack Is Equally Sensitive as PCR GeneScan and Sanger Sequencing for Detection of Clonal Rearrangements in ALL Patients"

_diagnostics, 2022, doi:10.3390/diagnostics12061389_

Round 1

Reviewer 1 Report

The authors evaluated the performance a NGS-based clonality detection assay in the context of acute lymphoblastic leukemia as compared to internationally approved gold standard protocol Biomed-2. As NGS becomes technically feasible in the vast majority of medical labs the topic is of great interest since clonality detection and clono-specific minimal residual disease measurement is a major endpoint in the therapeutic strategy of B and T-ALLs. The authors describe in details the workflow of LymphoTrack system including bioinformatic analysis. This point is critical since bioinformatics remains the Achilles’ Heel of many labs especially in the setting of clonality analysis. Quality control steps and validation criteria are also well described.

11-     As a quantitative threshold of 5% to 10% is required in the interpretation process, the authors should precise if the system allows the estimation of PCR duplicates, almost unavoidable using an amplicon-based enrichment protocol. Were control samples with known tumor burden included ?

22- Table 1 : the targets included into LymphoTrack PCR system as compared to Biomed-2 protocol should be briefly discussed regarding immunoglobulin heavy chain gene rearrangements: IgH FR1 only versus IgH-FR1+FR2+FR3 respectively (see also point 4)

33- As compared to Biomed-2 standardized multiplex PCR/Genescan approach the LymphoTrack system shows similar performance for the detection of Immunoglobulin and TCR genes rearrangements. The data presented here are convincing.

  4- As it stands, this system is dedicated to clonality detection. In section 4.2, the authors noticed that TCRD and incomplete IgH rearrangements are not currently available which may be a limitation for the design of clono-specific PCR protocols in the context of ALL. The percentage of such patients in which less than two informative clones were detected should be more clearly precised.

    5- The clinical presentation of ALLs may vary from overt leukemia to lymphoblastic lymphomas with no or low blastic infiltration. In this setting an estimation of the test limit of detection using cell dilutions would add decisive information.

5       

Author Response

The authors evaluated the performance a NGS-based clonality detection assay in the context of acute lymphoblastic leukemia as compared to internationally approved gold standard protocol Biomed-2. As NGS becomes technically feasible in the vast majority of medical labs the topic is of great interest since clonality detection and clono-specific minimal residual disease measurement is a major endpoint in the therapeutic strategy of B and T-ALLs. The authors describe in details the workflow of LymphoTrack system including bioinformatic analysis. This point is critical since bioinformatics remains the Achilles’ Heel of many labs especially in the setting of clonality analysis. Quality control steps and validation criteria are also well described.

1-    As a quantitative threshold of 5% to 10% is required in the interpretation process, the authors should precise if the system allows the estimation of PCR duplicates, almost unavoidable using an amplicon-based enrichment protocol. Were control samples with known tumor burden included ?

Answer: It would certainly be of interest to elucidate to what extent clonal rearrangements may be detected with LymphoTrack in healthy individuals. While we agree with the reviewer that it would be interesting to look at the ability of this technology to discriminate between reactive lymphoproliferation from leukemic clones, this was not the aim of this study. Our overall aim was to perform a technical validation of a commercially available kit compared to gold standard techniques such as fragment analysis, Sanger sequencing and RQ-PCR. If this assay had been a LDT (i.e. in house design as opposed to commercially designed kit), we agree that to assess the performance of the assay, serial dilution experiments using clonal control DNA such as polyclonal tonsil DNA followed by statistical determination of the assay limit of detection would have been required. This was not necessary for this assay since the kit has been commercially validated and pre-defined cut-off levels had already been determined using tonsil DNA during the internal validation process. We followed all manufacturers guidelines rigorously and adhered to all cut-off and detection levels specified. We have added text to the methods section that control DNA from tonsil was used as control material, and that all cutoffs used were based on the LymphoTrack guidelines.

2- Table 1 : the targets included into LymphoTrack PCR system as compared to Biomed-2 protocol should be briefly discussed regarding immunoglobulin heavy chain gene rearrangements: IgH FR1 only versus IgH-FR1+FR2+FR3 respectively (see also point 4)

Answer: We used the LymphoTrack single master mix IgH FR1, to amplify the entire target region (amplicons). In our experience of investigating IGH gene rearrangements in immature lymphoblasts compared to possibly hypermutated mature lymphoma cells, the use of FR2+FR3 is not necessary for our target detection since lymphoblasts are not hypermutated. Text has been added to the results section to address this matter.

3- As compared to Biomed-2 standardized multiplex PCR/Genescan approach the LymphoTrack system shows similar performance for the detection of Immunoglobulin and TCR genes rearrangements. The data presented here are convincing.

Answer: We thank the reviewer for agreeing with our conclusion.

4- As it stands, this system is dedicated to clonality detection. In section 4.2, the authors noticed that TCRD and incomplete IgH rearrangements are not currently available which may be a limitation for the design of clono-specific PCR protocols in the context of ALL. The percentage of such patients in which less than two informative clones were detected should be more clearly precised.

Answer: We agree with the reviewer and describe this issue in the discussion. We have added information to the discussion on the number of cases with only one informative clone.

  5- The clinical presentation of ALLs may vary from overt leukemia to lymphoblastic lymphomas with no or low blastic infiltration. In this setting an estimation of the test limit of detection using cell dilutions would add decisive information.

Answer: Indeed, when the blast count is less than 20 percent at the time of diagnosis it is not feasible to use the sample for adequate primer testing and follow-up MRD. The aim of this study was however to study patients with ALL (i.e. blast count above 20 percent) and compare the clonality pattern of LymphoTrack vs GeneScan for MRD target and not to evaluate Lymphotrack for MRD detection.

Reviewer 2 Report

Fine English revision 

Revise the methods in term of explanation of the techniques

Author Response

1. Fine English revision 

Answer: Minor English misspellings and grammatical errors have been corrected.

2. Revise the methods in term of explanation of the techniques

Answer: We have added information to the introduction that explains the techniques used in the study: Compared to the previous gold standard method of sanger sequencing, which sequences a single DNA fragment at a time, molecular diagnostics has moved towards next-generation sequencing (NGS) technology, the processing of millions of fragments simultaneously by deeper sequencing depth.

Reviewer 3 Report

The manuscript describes the comparison and validation of the NGS method for determining B and T-cell clonality in acute leukemia with the conventional approaches (GeneScan + Sanger sequencing). The research topic is important and relevant. The authors have analyzed a significant amount of material and make a reasonable conclusion about the validity of the NGS method for determining clonality in acute leukemia and subsequent selection of targets for MRD monitoring. The article may certainly be of interest to the readers of MDPI Diagnostics, however, some points need to be clarified.
1 High throughput sequencing of IG and T-cell receptor repertoire has been reported by numerous labs. It could be done by different approaches including NGS library preparations from DNA fragments amplified with Biomed, framework, or leader primers. Furthermore, appropriate software for processing raw NGS data and clonotype analysis is freely distributed for noncommercial use (https://github.com/milaboratory/mixcr). It would be worthwhile to compare sensitivity and specificity of diverse approaches rather than promoting single commercial reagent kit from InVivoScribe.  2 No control sample (healthy donors, patients with reactive lymphoproliferation, etc.) included in the study. It is well known, that clonal expansion (especially for T-cell clones) could be found in non-malignant conditions. It is interesting to demonstrate whether proposed criteria (10% in ≥20000 reads and 5% in 10000-20000 reads) may discriminate leukemic clone from reactive. 3 What was the percentage of productive vs nonproductive clonal rearrangements in the studied sample? Probably supporting raw data as supplementary material could be beneficial. 4 In some cases Sanger sequencing may not resolve rearranged sequences due to more than one clone amplified with same primer pair. These cases do require NGS approach for MRD target design. Do authors came across such occasions in the sample studied? 

Author Response

The manuscript describes the comparison and validation of the NGS method for determining B and T-cell clonality in acute leukemia with the conventional approaches (GeneScan + Sanger sequencing). The research topic is important and relevant. The authors have analyzed a significant amount of material and make a reasonable conclusion about the validity of the NGS method for determining clonality in acute leukemia and subsequent selection of targets for MRD monitoring. The article may certainly be of interest to the readers of MDPI Diagnostics, however, some points need to be clarified.

  1. High throughput sequencing of IG and T-cell receptor repertoire has been reported by numerous labs. It could be done by different approaches including NGS library preparations from DNA fragments amplified with Biomed, framework, or leader primers. Furthermore, appropriate software for processing raw NGS data and clonotype analysis is freely distributed for noncommercial use (https://github.com/milaboratory/mixcr). It would be worthwhile to compare sensitivity and specificity of diverse approaches rather than promoting single commercial reagent kit from InVivoScribe. 

Answer: We thank the reviewer for this excellent suggestion. As mentioned by the reviewer, there are numerous different approaches available to analyze rearrangements with high throughput sequencing and it would be interesting to investigate different approaches. However, our study was not aimed at providing a comprehensive review of all methodologies and softwares available, instead the scope of this manuscript was to compare PCR GeneScan with LymphoTrack using IdentiClone Clonality Assays (InVivoScribe), since InVivoScribe is the assay used in our lab in the clinical setting, which is why, we focused on the cut-offs and software validated and recommended by InVivoScribe .

  1. No control sample (healthy donors, patients with reactive lymphoproliferation, etc.) included in the study. It is well known, that clonal expansion (especially for T-cell clones) could be found in non-malignant conditions. It is interesting to demonstrate whether proposed criteria (10% in ≥20000 reads and 5% in 10000-20000 reads) may discriminate leukemic clone from reactive.

Answer: It would certainly be of interest to elucidate to what extent clonal rearrangements may be detected with LymphoTrack in healthy individuals. While we agree with the reviewer that it would be interesting to look at the ability of this technology to discriminate between reactive lymphoproliferation from leukemic clones, this was not the aim of this study. Our overall aim was to perform a technical validation of a commercially available kit compared to gold standard techniques such as fragment analysis, Sanger sequencing and RQ-PCR. If this assay had been a LDT (i.e. in house design as opposed to commercially designed kit), we agree that to assess the performance of the assay, serial dilution experiments using clonal control DNA such as polyclonal tonsil DNA followed by statistical determination of the assay limit of detection would have been required. This was not necessary for this assay since the kit has been commercially validated and pre-defined cut-off levels had already been determined using tonsil DNA during the internal validation process. We followed all manufacturers guidelines rigorously and adhered to all cut-off and detection levels specified. . We have added text to the methods section that control DNA from tonsil was used as control material, and that all cutoffs used were based on the LymphoTrack guidelines.

  1. What was the percentage of productive vs nonproductive clonal rearrangements in the studied sample? Probably supporting raw data as supplementary material could be beneficial.

Answer: When identifying clonal rearrangements for possible MRD targets in acute leukemia we do not consider whether the rearrangement is productive or non-productive (as we do in somatic hypermutation analysis for CLL), since the overall aim is to identify rearrangements appropriate for MRD target and thus it is of less importance whether the rearrangement is productive or not. The datasets are available on reasonable request, as stated in the manuscript.

  1. In some cases Sanger sequencing may not resolve rearranged sequences due to more than one clone amplified with same primer pair. These cases do require NGS approach for MRD target design. Do authors came across such occasions in the sample studied?

Answer: The reviewer is correct in stating that more than one clone may be amplified by a primer pair and we did indeed observe such cases. For these cases we would have had to perform cloning of PCR products followed by subsequent Sanger sequencing, however, a major advantage of NGS is that it is no longer necessary to perform laborious and time-consuming “cloning” procedures. . A sentence to this effect has been added to section 3.6 of the manuscript. i.e.  3.6. Ambiguous rearrangements.

Round 2

Reviewer 3 Report

No comments.